# Use of a GP-endorsed 12 months' reminder letter to promote uptake of bowel scope screening: protocol for a randomised controlled trial in a hard-to-reach population

Christian von Wagner,[1] Yasemin Hirst,[1] Sara Tookey,[1] Robert S. Kerrison,[1] Sarah Marshall,[2] Andrew Prentice,[2] Daniel Vulkan,[3] Una Macleod,[4] Stephen Duffy[3]

[1]Research Department of Behavioural Science and Health, University College London, London, UK
[2]St Mark's Bowel Cancer Screening Centre, London North West Healthcare NHS Trust, Harrow, UK
[3]Policy Research Unit in Cancer Awareness, Screening and Early Diagnosis, Wolfson Institute of Preventive Medicine, Barts and The London School of Medicine and Dentistry, Queen Mary University of London, London, UK
[4]Centre for Health and Population Sciences, Hull York Medical School, University of York, Heslington, UK

**Correspondence to**
Dr Christian von Wagner; c.wagner@ucl.ac.uk

## ABSTRACT

**Introduction** Flexible sigmoidoscopy (FS) screening is associated with reduced colorectal cancer incidence and mortality when offered as a one-off test to men and women aged 55–64. The test, also referred to as the 'bowel scope screening' (BSS) test, was added to England's national Bowel Cancer Screening Programme in March 2013, where it is offered to men and women aged 55. Since its implementation, uptake of the BSS test has been low, with only 43% of the eligible population attending an appointment. Sending non-participants a reminder at age 56 has been shown to improve uptake by up to nine percentage points at a single centre in London; we hypothesise that adding a general practitioners (GPs) endorsement to the reminder could improve uptake even further.

**Methods and analysis** This paper describes the protocol for a randomised controlled trial which will examine the effectiveness of adding a GPs endorsement to a reminder for BSS non-participants aged 56. All screening-eligible adults who have not responded to a BSS appointment at London North West Healthcare NHS Trust within 12 months of their initial invitation will be randomised to receive either a GP-endorsed reminder letter or reminder letter without GP endorsement. The primary outcome will be the proportion of individuals screened within each group 8 weeks after the reminder. Statistical comparisons will be made using univariate and multivariate logistic regression, with 'uptake' as the outcome variable, GP reminder group as the exposure and sociodemographic variables as covariates.

**Ethics and dissemination** The study was approved by the Yorkshire & Humber—Bradford Leeds Research Ethics Committee (16/YH/0298) and the Confidentiality Advisory Group (17/CAG/0162). The results will be disseminated in a peer-reviewed journal in accordance with the Consort statement and will be made available to the public.

**Trial registration number** ISRCTN82867861

## Strengths and limitations of the study

► If effective, this study will provide a cost-effective means to promote uptake of flexible sigmoidoscopy (FS) screening in an organised programme.

► The key importance of this randomised controlled trial protocol is that it highlights a methodology for delivering general practitioner (GP)-endorsed reminders to examine whether a 12 months' reminder letter can be further enhanced to improve screening uptake in an organised FS-based screening programme for colorectal cancer.

► There is a strong public health mandate to develop system-friendly interventions to promote uptake of the bowel scope screening programme in England, particularly among socioeconomically deprived groups, where the potential for improving uptake is greatest.

► The study population is limited to the patients who are registered at the participating GP practices in London Boroughs of Hillingdon, Harrow and Brent.

early, when it is more treatable.[3–6] Screening can additionally prevent cases through the early detection and removal of adenomas: the precancerous lesions from which most CRCs develop.[7–9]

Randomised controlled trials (RCTs) examining the effectiveness of flexible sigmoidoscopy (FS) screening to improve CRC outcomes highlight that disease-specific mortality and incidence are reduced by 32% and 50%, respectively, when the test is performed just once between the ages of 55 and 64.[8 10–12] On this basis, the National Health Service (NHS) in England added 'once-only' FS screening for men and women aged 55 (known as bowel scope screening, BSS) to the National Bowel Cancer Screening Programme (BCSP) in March 2013.[13] Since that time, uptake of FS in the English BCSP

## INTRODUCTION

Colorectal cancer (CRC) is the fourth most common cancer in the UK and the second leading cause of cancer deaths.[1 2] Screening is able to improve survival by detecting CRC

has been only 43%,[14] which is considerably lower than the uptake for the faecal occult blood test (FOBt) which is offered biennially to those aged 60–74, and dramatically lower than the rates observed for breast and cervical screening (54%, 76%, 78%, respectively,[15–17]).

Barriers to FS screening include practical barriers (ie, inconvenient appointment time, difficulties travelling to the appointment and so on), worry about pain, discomfort or injury associated with the examination, and not wanting to know about any health issues.[18] Several studies attempting to address these barriers have examined the use of self-referral reminders (reminders which prompt former non-participants to self-refer for screening) 12 and 24 months after their initial invitation.[19–21] These have been shown to facilitate uptake in as much as 21.5% of former non-participants. While these results are highly promising, the annual reminder concept is still relatively unexplored, and there is considerable scope for modification and refinement of the intervention content. For example, there is now considerable evidence that a general practitioner (GP) endorsement of cancer screening is positively associated with uptake.[22–24] Previous research examining the relative importance of barriers to CRC screening indicate that primary care should play a key role in encouraging uptake, highlighting that the two most important barriers to CRC screening among individuals who have never been screened are 'failure of the clinician to suggest screening' and 'not knowing testing was necessary'.[25]

A recent study conducted in Australia demonstrated that a theory-based modification to the advanced notification letter improved uptake among men and was highly cost-effective.[22] A recent review has found that the inclusion of a GP's endorsement on the invitation letter can improve uptake of CRC screening with the FOBt; however, none of the studies identified in the review used FS screening uptake as an outcome.[26] This paper describes the protocol for an RCT which will examine the effectiveness of adding a GP endorsement to a reminder sent to BSS non-responders 12 months after their initial invitation.

## AIMS
The primary aim of this RCT will be to test whether adding a GP endorsement to the 12 months' reminder letter improves the uptake of BSS among previous non-responders over and above a 12-month reminder letter without a GP endorsement. The secondary aim will be to examine demographic differences in uptake in response to the GP-endorsed reminder letter and the standard annual reminder letter (ie, a reminder without a GP endorsement).

## METHODS AND ANALYSIS
### Study design
This study will be a non-clinical RCT with two parallel arms (see figure 1). The intervention group will receive a GP-endorsed reminder letter (online supplementary appendix A) by post, the control group will receive the same reminder letter by post, minus the GP endorsement (online supplementary appendix B).

### Study setting
This RCT will be conducted in London, England, at London North West Healthcare NHS Trust in Harrow in Summer 2018.

### Eligibility criteria
Adults will be eligible to take part in the study if they: (1) are aged 56 years at the time of enrolment, (2) are registered with a consenting general practice served by the BSS centre at London North West Healthcare NHS Trust, (3) have been offered but not responded to a routine BSS appointment for at least 12 months, (4) meet the clinical eligibility criteria for BSS and (5) have not opted out from sharing their personal data for purposes beyond direct care (from here on referred to as type 2 objectors).

NHS Digital, formerly known as the Health and Social Care Information Centre, will be responsible for identifying potentially eligible adults and excluding any individuals who do not meet the criteria for the study. NHS Digital will exclude individuals if they: (1) have an open episode status, for example, postponed the appointment to another date; (2) contacted the screening centre and declined; (3) contacted the screening programme and were deemed medically unfit; (4) contacted the screening programme to confirm they would be attending their appointment, but then did not attend.

### Blinding and randomisation
Adults who are eligible for inclusion will be entered into a computerised study database and randomised (in a 1:1 ratio) to receive either a 'standard reminder letter' or a 'GP-endorsed reminder letter' 12 months after their initial invitation using simple pseudorandom allocation methods.

### Preintervention phase: recruitment of practices
We will send invitations to GP practices which are being served by London North West Healthcare NHS Trust and have been included in the BSS programme for at least 12 months. Consenting GP practices will be permitting their practice name to be included in the GP-endorsed reminder letter. Using a recruitment strategy similar to a recent RCT with GP practices in London, we are expecting approximately 50% of GP practices to sign up using a strategy of email invitations and weekly reminders.[27]

### Intervention phase: reminder delivery
#### The 12 months' reminder letter (standard reminder group)
The standard reminder letter will be the same reminder letter that was used in the previous trials (see online supplementary appendix A[19–21]). It will be a personally addressed letter from BSS centre at the North West London Hospitals Trust that will invite recipients to make an appointment by returning an 'appointment-request slip' or calling the Freephone number at the screening centre. As with the previous

von Wagner C, et al. BMJ Open 2018;8:e022263. doi:10.1136/bmjopen-2018-022263

trials, the reminder will also give recipients the option to express a preference for the day and time of their appointment.

### The 12 months' reminder letter with GP endorsement (intervention group)

The GP-endorsed reminder letter will be the same as the standard reminder letter, except that it will contain an additional statement of GP endorsement which states: 'Your GP practice, xxxx xxxxx, supports the NHS Bowel Scope Screening Programme' (see online supplementary appendix B).

### Timeline

Twelve months after receiving an initial invitation for BSS, all eligible adults will receive a reminder letter (either the endorsed or standard letter) with the option of indicating preferred days and times to schedule their BSS appointment at the screening centre based at London North West Healthcare NHS Trust.

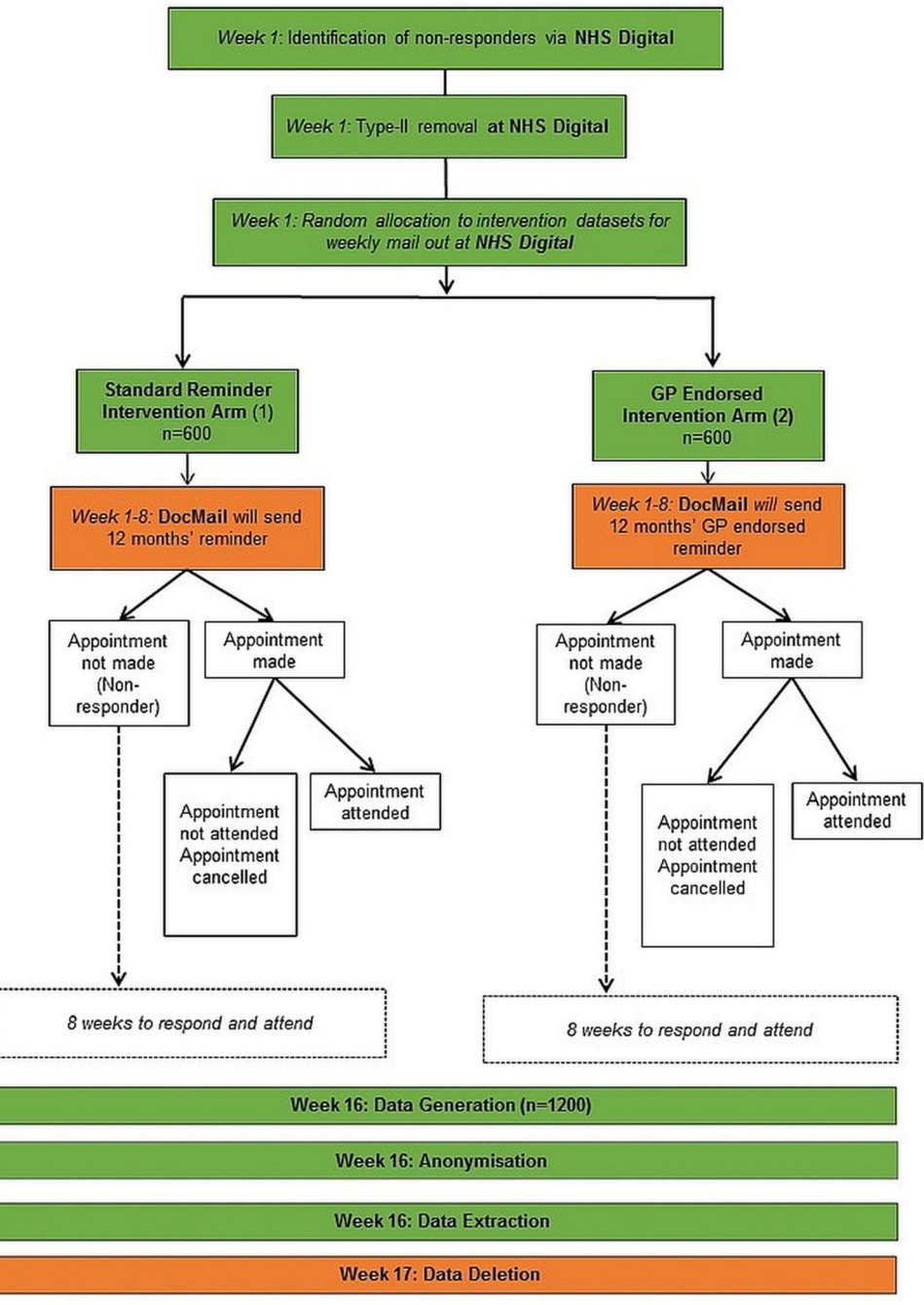

**Figure 1** GP-endorsed reminders Consort flow diagram. GP, general practitioner; NHS, National Health Service.

## Sample size calculation

The study has been designed to detect a five percentage point increase in uptake between the GP-endorsed and standard reminder group. As a GP-endorsed reminder has not previously been tested in the context of an organised FS screening programme for CRC, the estimates for this study were based on the effect size of using GP endorsement to promote uptake of screening for CRC using alternative tests (such as the FOBt[22–24]). Hewitson and colleagues[24] found that the response rate of individuals receiving usual care, versus a GP-endorsed invitation letter, increased the rate of response by 5%, from 10% (as observed for a standard reminder in previous work) to 15%. To detect this increase in uptake with 80% power at the 5% alpha level, with two-sided testing, approximately 600 participants per trial arm are required, giving a total sample size of 1200 participants.

## Data processing and data collection

1. Identification: Adults will be identified by NHS Digital only if they are registered with a consenting GP practice. Identification of eligible participants will take place using data contained within the Bowel Cancer Screening System which provides an up-to-date electronic record of uptake data for individuals enrolled in the national BSS.

2. Data cleanse: NHS Digital will query the data of non-participants to identify type 2 opt-outs. Using the cleansed dataset, the first 1200 individuals will be selected to be included in the RCT database.

3. Randomisation: NHS Digital will randomise eligible participants using the prerandomised dataset. The database will be split into two separate files with an equal number of people in each document using the study groups.

4. Intervention: NHS Digital will share the data associated with each mail-out for the intervention and the control group with the mailing company called CFH Docmail Limited. CFH Docmail is an NHS Information Governance toolkit accredited mailing company that will be facilitating the mail merge throughout the study. The dataset will include study group, unique ID, personal contact details, NHS number and the practice name (only for intervention group). Reminders will be produced by merging each study group database with the reminder letter templates. The full package will include: the reminder letter, standard information booklet and a freepost return envelope. This process of sending previous non-participants a reminder 12 months after their initial invitation will continue until the study sample size (n=1200) is reached. We plan to send out 150 letters per week and send out all the reminders in 8 weeks.

5. Data generation: once letters are sent, we will allow 8 weeks for all individuals to respond to the self-referral reminder. At the 16th week of the trial, NHS Digital will query the screening episode status of all individuals using their NHS numbers. If previous non-responders did not respond, their status will not change, and they will remain 'non-responders'. If people confirmed and attended their appointment, they will be coded as 'attended'. If people contacted the centre and cancelled their appointment and did not book another appointment, they will be considered 'decliners'. If someone confirmed an appointment but did not attend, they will be coded as 'non-attenders'. We will also request a separate category for those who have contacted the screening centre, had the initial confirmation and attended the clinic but not eligible to have the full investigation. People will be adequately screened, if they have had the full bowel scope investigation and received a definitive clinical result. For this purpose, we will additionally request the screening outcome of those who had the screening, and also information about the follow-up colonoscopy investigations. As a result, we will be able to distinguish between: (1) those who reacted to the self-referral reminders (responders) versus those who did not (non-responders), (2) those who successfully attended screening (attenders) versus those who did not (non-attenders) and (3) those who were adequately screened versus those who were not. In addition, NHS Digital will provide the Index of Multiple Deprivation (IMD) scores, using the postcodes of each individual included in the study.

6. Data extraction: At the end of the study, after all data have been collated, the study database will be duplicated by NHS Digital. One copy will be anonymised for analysis by the University College London (UCL) research team. In the anonymisation process, all identifiable information including NHS number, full name, address, GP code and GP name will be excluded.

7. Data deletion: Once the anonymised dataset is sent to the research team at UCL and verified, the research team will ask CFH Docmail to destroy all the datasets that were shared by NHS Digital. Only the research team at UCL will have access to the final dataset.

## Primary outcome

The primary outcome of this study will be the proportion of individuals attending a BSS appointment within each group. Uptake will be determined by checking the episode status of each individual included in the study 8 weeks after the distribution of the final reminder letter.

## Statistical analysis

The primary outcome will be analysed to test for a significant difference in uptake between the two groups using univariate and multivariate logistic regression, with 'adequately screened' as the outcome measure, 'trial arm' as the exposure and 'gender' and IMD as the covariates. Secondary analyses will look at whether there are differences in sociodemographic characteristics of non-responders, decliners and non-attenders. Sample characteristics will be reported using descriptive statistics.

The comparison of overall uptake between trial arms will be presented using ORs and 95% CIs.

## Ethics and dissemination

The timeliness and the feasibility of the study was approved by the Public Health England Bowel Cancer Screening Programme Research Advisory Committee (ID_192). The study was also submitted to the Confidentiality Advisory Group (17/CAG/0162) for Section 251 exemption of the NHS Act 2006 which permits individual data to be processed without consent when the reasons for no consent is justified and the proposed study is in the public interest. However, the project website is designed to inform members of the public who will be included in the trial, timelines, objectives and how their data will be processed (http://www.ucl.ac.uk/iehc/research/behavioural-science-health/research/cancer-communication-screening/gp-reminders-for-bss-non-participants). As part of this patient and public notification, individuals are given the opportunity to opt-out in line with the CAG requirements. The explicit consent was not deemed necessary because the annual reminders are in part being sent as routine practice at the BSS centre at London North West Healthcare NHS Trust as part of a Commissioning for Quality and Innovation awarded by NHS England in 2017, and at the end of the project, the research team at UCL will not be receiving identifiable information. The exemption was necessary for the secure and fair data processing between NHS Digital and CFH Docmail. A Section 251 exemption was permitted to the CFH Docmail to receive the name, address and the NHS number of the non-responders of BSS from NHS Digital for the duration of the study and subsequently after the annual reminders.

We will disseminate the outcomes of the study using academic publications, the study website, social media and also send a report to the GP practices that were invited to the study once the results are published. We will aim to publish the results in a peer-reviewed journal in accordance with the CONSORT statement. We will also use the study website to inform the public about the study outcomes (http://www.ucl.ac.uk/iehc/research/behavioural-science-health/research/cancer-communication-screening/gp-reminders-for-bss-non-participants).

## Patient and public involvement

Members of the public were involved in the design of the self-referral reminder letter used in this study and those conducted before it.[19–21] Information on the study website and the opt-out form were evaluated and informed by a patient representative. To make our findings more accessible to the public, we plan to report the outcomes of the trial on the study website in lay terms after the results are published in a peer-reviewed journal.

**Acknowledgements** We would like to acknowledge Josephine Ruwende and Jean-Pierre Laake for their advice and assistance in the recruitment of GPs for this study. We thank Lindy Berkman, our patient representative, for her support in the development of the information provided to the public through the study website and included in the opt-out forms.

**Contributors** CvW and RK conceived of the study. ST, YH, CvW and RK participated in its design. ST, YH, CvW, RK and ST wrote the first draft of the manuscript. RK and YH participated in the power calculations and statistical analysis. DV, AP, SM, SD and UM were involved in revisions of the manuscript. All authors read and approved the final manuscript.

**Funding** This work is part of the programme of the Policy Research Unit in Cancer Awareness, Screening and Early Diagnosis, which receives funding for a research programme from the Department of Health Policy Research Programme (106/001). It is a collaboration between researchers from seven institutions (Queen Mary University of London, University College London, King's College London, London School of Hygiene and Tropical Medicine, Hull York Medical School, Durham University and Peninsula Medical School).

**Disclaimer** The funder has played no role in study design and would have no influence in data collection, management, analysis and interpretation or any decisions about whether and where to publish the results of the study.

**Competing interests** None declared.

**Patient consent** Not requried.

**Ethics approval** The Yorkshire & Humber - Bradford Leeds Research Ethics Committee (16/YH/0298).

**Provenance and peer review** Not commissioned; externally peer reviewed.

**Data sharing statement** The data that support the findings of this study are available from the Bowel Cancer Research Committee, but restrictions apply to the availability of these data, which are used under licence for the current study, and so are not publicly available. Only the research team at UCL will have access to the final dataset. Anonymised data can be made available from the authors on reasonable request and with permission of the Bowel Cancer Research Advisory Committee after an application is made to the Office for Data Release for a data sharing agreement.

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
