## [Reviewer comments · BMJ Open]

ARTICLE DETAILS

TITLE (PROVISIONAL)	Use of a GP-endorsed 12 months' reminder letter to promote uptake of bowel scope screening: Protocol for a randomised controlled trial in a hard-to-reach population.
AUTHORS	von Wagner, Christian; Hirst, Yasemin; Tookey, Sara; Kerrison, Robert; Marshall, Sarah; Prentice, Andrew; Vulkan, Daniel; Macleod, Una; duffy, stephen

VERSION 1 – REVIEW

REVIEWER	Carlo Senore SSD Epidemiologia e Screening - CPO, AOU Città della salute e della scienza, Turin, Italy
REVIEW RETURNED	05-Mar-2018

GENERAL COMMENTS	The authors are presenting the protocol of a well designed study, addressing a relevant issue. The results of the trial may be directly transferred to the screening practice. Participation rate is generally lower with CRC screening, as compared to breast or cervical cancer screening. Availability of several effective tests may favour uptake, as long as patients can be offered different options, but available evidence about effective methods to increase uptake, in particular for a test like FS, which was introduced only recently is limited. Also, as long as the test is effective when performed once in the life-time, new approaches to promote participation need to be developed, as cumulative, rather than repeated participation, is relevant. The trial was designed to evaluate one possible strategy to increase the uptake, by favouring the response at 1 year of those who refused the initial invitation. The study design seems adequate, both in terms of randomisation and outcome ascertainment procedures and in terms of sample size. The methodology adopted for the analysis is appropriate.
--

REVIEWER	Jaroslaw Regula Department of Gastroenterology, Maria Sklodowska-Curie Institute - Cancer Centre, Warsaw, Poland
REVIEW RETURNED	06-Mar-2018

GENERAL COMMENTS	This is a manuscript describing methodology of a randomized controlled trial assessing the role of GP recommendation added to a routine invitation for sigmoidoscopy screening examination. It is a valuable paper and I suggest publishing it mainly due to very
---

	detailed description of methodology. I have the following concern/remark. Despite the intention of authors to have two identical groups of invitees – it will probably be not possible with current flow of the study. Intervention group will consist of people whose GP's agreed to participate in the study. The control group however – if I well understand the protocol - will consist of people whose GP's will not be asked to participate in the study. I think people for both group should be recruited from those whose GP's agreed to be a part of the study irrespective whether their patients will be in intervention group or control group. See text, page 5: "The coverage of GP endorsement will be limited to those areas served by GPs willing to participate in our trial". I would prefer the situation that "the study (with or without endorsement) will be conducted in the areas served by GPs willing to participate in our trial".
--	---

VERSION 1 – AUTHOR RESPONSE

Editorial Requests:

- Re SPIRIT Item 27: please clarify where in the paper it specifically states how personal information about potential and enrolled participants will be collected, shared, and maintained in order to protect confidentiality before, during, and after the trial.

In the sections titled "data processing and data collection", and "ethics and dissemination", we describe in detail how we will protect confidentiality before, during and the after the trial. Due to a very detailed description, we mentioned both pages 10-11 in the spirit checklist.

- Re SPIRIT Item 29: can you please also clarify where in the paper it states who will have access to the final trial dataset, or revise your manuscript accordingly?

Page 13 on our submission & page 14 on the pdf version of the manuscript includes the data sharing statement. To clarify who will have access to the final trial dataset, we included a sentence that states that only the research team at UCL will have access to the final dataset in the data sharing statement and also in page 10 where we describe the data deletion process.

Reviewer: 2

Reviewer Name: Jaroslaw Regula

Institution and Country: Department of Gastroenterology, Maria Sklodowska-Curie Institute - Cancer Centre, Warsaw, Poland
Competing Interests: None

This is a manuscript describing methodology of a randomized controlled trial assessing the role of GP recommendation added to a routine invitation for sigmoidoscopy screening examination. It is a valuable paper and I suggest publishing it mainly due to very detailed description of methodology.

I have the following concern/remark. Despite the intention of authors to have two identical groups of invitees – it will probably be not possible with current flow of the study. Intervention group will consist of people whose GP's agreed to participate in the study. The control group however – if I well understand the protocol - will consist of people whose GP's will not be asked to participate in the study. I think people for both group should be recruited from those whose GP's agreed to be a part of the study irrespective whether their patients will be in intervention group or control group.

See text, page 5: “The coverage of GP endorsement will be limited to those areas served by GPs willing to participate in our trial”. I would prefer the situation that “the study (with or without endorsement) will be conducted in the areas served by GPs willing to participate in our trial”.

We apologise if our statement on page 5 was not clear. We can confirm that individuals enrolled in both arms of the trial will be included from the GP practices that will consent to take part in the study. We thought this was clear in our eligibility criteria on page 6. However, we agree with the reviewer that this statement could be confusing and we think that the reworded version as per the reviewer suggestion would have the same meaning as bullet point 4. Thus, we decided to delete the last bullet point on page 5. We included a statement on page 8, under data processing and data collection indicating that NHS digital will only identify people who are registered with a consenting GP practice.

FORMATTING AMENDMENTS (if any)

Required amendments will be listed here; please include these changes in your revised version:

- Kindly re-upload FIGURES in either TIFF or JPG format with at least 300 dpi resolution.

This is completed as requested.

- Please include Figure legends at the end of your main manuscript.

This is completed as requested.

- Kindly re-upload APPENDICES in PDF format.

This is completed as requested.

- Please mention the author (Macleod, Una) in your CONTRIBUTOR SHIP STATEMENT along with specific contribution/participation for the article.

Many thanks for highlighting this. We now amended the contributor statement to include Prof Macleod's contribution in the article.